

# LINC00106/RPS19BP1/p53 axis promotes the proliferation and migration of human prostate cancer cells

Lingxiang Lu[1,*], Zhen Tian[2,*], Jicheng Lu[3], Minjun Jiang[1], Jianchun Chen[1], Shuai Guo[1] and Yuhua Huang[2]

[1] Department of Urinary Surgery, Suzhou Ninth People's Hospital, Soochow University, Suzhou, Jiangsu, China
[2] Department of Urinary Surgery, The First Affiliated Hospital of Soochow University, Suzhou, Jiangsu, China
[3] Oncology Department, Suzhou Ninth People's Hospital, Soochow University, Suzhou, Jiangsu, China
[*] These authors contributed equally to this work.

Corresponding authors
Shuai Guo, mnwk639@163.com
Yuhua Huang, sdfyy_hyh@163.com

## ABSTRACT

**Background**. Prostate cancer (PCa) is among the most prevalent cancers in males with high biochemical recurrence risk. LINC00106 contributes to the carcinogenesis of Hepatocellular carcinoma (HCC). However, it is unclear how it affects PCa advancement. Here, we studied LINC00106's effects on PCa cells' ability to proliferate, invade, and metastasize.
**Methods**. The data of LINC00106 from The Cancer Genome Atlas (TCGA) in human PCa tissues were analyzed using TANRIC and survival analysis. In order to determine the expression levels of genes and proteins, we also performed reverse transcription-quantitative PCR and western blot analysis. The migration, invasion, colony formation, and proliferation (CCK-8) of PCa cells with LINC00106 knockdown were investigated. The impact of LINC00106 on cell proliferation and invasion was also analyzed in mice. LncRNA prediction software catRAPID omics v2.1 (catRAPID omics v2.0 (tartaglialab.com)) was used to predict proteins that might interact with LINC00106. The interactions were verified via RNA immunoprecipitation and RNA pull-down assays and finally, the interaction between LINC00106 and its target protein and the p53 signaling pathway was studied using a dual-luciferase reporter assay.
**Results**. In PCa, LINC00106 was over-expressed in comparison to normal tissues, and it was linked to an unfavorableprognosis. *In vitro* and *in vivo* analyses showed that downregulating LINC00106 decreased PCa cells'ability to proliferate and migrate. A common regulatory axis generated by LINC00106 and RPS19BP1 prevents p53 activity.
**Conclusion**. Our experimental data indicate that LINC00106 functions as an oncogene in the onset of PCa, and the LINC00106/RPS19BP1/P53 axis canserve as a novel therapeutic target for PCa treatment.

# INTRODUCTION

With over 1.4 million new cases globally and 375,000 fatalities recorded in 2020, prostate cancer (PCa) remains one of the major factors contributing to morbidity and mortality in men (*Bray et al., 2018*; *Schatten, 2018*). Despite the rising PCa incidence rate, the only

known risk factors for it are age, race, and family history of PCa (*Ha Chung, Horie & Chiong, 2019*). Although clinical parameters like prostate-specific antigen, radiographic diagnosis, and histopathological scores (Gleason score) can be used for risk stratification, they do not achieve accurate treatment outcomes in patients (*Tosoian et al., 2017*). Therefore, unnecessary treatment, not administering potentially beneficial treatment, and toxicity remain unaddressed challenges (*Schröder et al., 2014*). Current treatments for PCa include anti-androgen therapy, chemotherapy, and radiotherapy, which have reportedly improved the survival rate of patients significantly (*Litwin & Tan, 2017*; *Nader, Amm & Aragon-Ching, 2018*; *Kamran & D'Amico, 2020*). However, drug resistance, recurrence, and metastasis of PCa and other problems have reduced the benefits of these treatment modalities (*Yamada & Beltran, 2021*; *Sartor & De Bono, 2018*). Reliable biomarkers are needed to facilitate decision-making in challenging clinical settings. For patients with potentially indolent low-risk PCa, current European cancer guidelines have outlined various evidence-based treatment options (*Cech & Steitz, 2014*), and validated biomarkers are available to help with treatment decision-making. Therefore, novel biomarkers will be of great clinical value if they achieve favorable treatment outcomes in patients with advanced PCa.

Noncoding RNAs (ncRNAs) exert significant regulatory functions in various disease pathologies, especially cancer (*Memczak et al., 2013*). The ncRNAs include microRNAs, repetitive RNAs, intronic RNAs, and long ncRNAs, a diverse group of biomolecules with broad potential to control gene expression (*Matsui & Corey, 2017*). Many ncRNAs are abnormally expressed in cancer and drive tumorigenesis or progression (*Bisogno & Keene, 2018*). As a newly discovered lncRNA, the impact of LINC00106 on cancer progression is still not well studied. Compared to healthy liver tissue, hepatocellular carcinoma (HCC) tissues exhibit upregulation of LINC00106 expression. In individuals with HCC patients, its level was related to tumor stage, lymphatic metastasis, and overall survival (*Liang et al., 2021*). METTL3 specifically maintains LINC00106 stability by promoting the differentiation of m6A in liver cancer cells, increasing the abundance of LINC00106 in the nucleus, and promoting the stem cell and metastasis characteristics of liver cancer (*Liang et al., 2021*). Still, there is no information regarding the mechanism underlying the effect of LINC00106 in PCa.

The current study was conducted to better understand the function of LINC00106 in PCa. It was found that an unfavorable prognosis in PCa was associated with an elevated expression level of LINC00106. Additionally, we found that LINC00106 was expressed in PCa cell lines and significantly impacted their migration and proliferation. After that, an RNA-binding protein immunoprecipitation assay was performed to detect the pathways regulated by LINC00106. Finally, it can be concluded that LINC00106 plays a carcinogenic role in the occurrence and development of PCa.

## MATERIALS & METHODS

### Bioinformatics analysis

TANRIC, an open-access tool for the interactive study of lncRNAs in cancer, was used to analyze the data for LINC00106 in the TCGA of human PCa tissues (*Li et al., 2015*).

TANRIC integrates data from TCGA, CCLE, and other large-scale tumor research projects to analyze lncRNA expression levels in various tumors and provide differential analysis data and data on the correlation of lncRNA with clinical information and genomic data (*Cancer Genome Atlas Research Network et al., 2013*; *Tomczak, Czerwińska & Wiznerowicz, 2015*; *Nusinow et al., 2020*). LncRNAs in the Gencode database (*Harrow et al., 2012*) were used as the standard for analysis, and the ones that overlapped with the exons of protein-coding genes were filtered out before analysis. RNA-seq data of TCGA, CCLE, and other projects were downloaded for lncRNA quantification in tumors using the quantitative method of reads per kilobase of transcript per million mapped reads. LncRNAs with an expression level greater than 0.3 in all samples were screened for subsequent differential analysis. Download sample-related clinical information and prognosis from TCGA (https://xenabrowser.net/datapages/) for correlation analysis between lncRNA and these data. Spearman correlation analysis was performed, with a correlation coefficient of 0.6 as the threshold.

## RNA extraction and reverse transcription-quantitative PCR

The total RNA from cells was extracted using the TRIzol reagent (Invitrogen, CA, USA). Using PrimeScript RT reagent (TaKaRa, Shanghai, China), cDNA was prepared by reverse transcription of extracted RNA following the protocol. SYBR Green Premix Ex Taq performed qPCR using a 7500 real-time PCR apparatus (TaKaRa, Shanghai, China). The levels of gene expression were normalized to 18s rRNA. The primer sequences are detailed as follows:

LINC00106, 5′-CCAGTGGTCACCTGAGATGG-3′(forward) and 5′-AGGACACCGTCT GTCTTACG-3′(reverse). RPS19BP1, 5′-AAGTCGGCACTGGACGAGTA-3′(forward) and 5′-TTCTGGCGCAAAATCTGCTG-3′(reverse). 18s rRNA, 5′-AAACGGCTACCACATCCAAG-3′(forward) and 5′-CCTCCAATGGATCCTCGTTA-3′(reverse).

## Cell culture and transfection

DU145 and PC3 cell lines were procured from the Institute of Cell Biology, affiliated with the Chinese Academy of Sciences (Shanghai, China). RPMI-1640 medium was used for culturing these cell lines. The overall conditions were set to be humid at 37 °C and 5% $CO^2$, supplemented with fetal bovine serum (10%). DU145 and PC3 cells were transfected with small interfering RNAs (siRNAs), including LINC00106, RPS19BP1, and siRNA (si-NC) as a negative control using Lipofectamine 2000 (Invitrogen, Waltham, MA, USA). The following siRNA sequences were used in our study:

si-LINC00106-1#: 5′-UUUUUUUUUUACAAUCACACAU-3′,
si-LINC00106-2#: 5′-UUUUUUUUUUUUUUACAAUCAC-3′,
si-RPS19BP1-1#: 5′-AGUUUCUGGGGCCUGAAUUGCC-3′,
si-RPS19BP1-2#: 5′-AAACUUCAGGUUUACUCUGAG-3′,
si-NC: 5′-ACGUGACACGUUCGGAGAA-3′.

GenePharma (Shanghai, China) synthesized the pcDNA3.1-RPS19BP1 and empty vector, which was transfected into DU145 and PC3 cells using Lipofectamine 2000.

## Cell viability and migration analysis

The 96-well plates were used to inoculate the transfected cells. The cell counting kit-8 (CCK8) test (Nantong Beyotime, China) was utilized to determine cell viability every 24 hrs. For the colony formation experiment, cells were inoculated into 6-well plates (500 cells/well), and incubated for two weeks, followed by methanol-assisted fixation for 30 min, dying with Beyotime 0.1% for 30 min, and then drying. Finally, the total number of cells was then determined. Cell migration was studied by Transwell chamber assay (BD Biosciences, Franklin Lakes, NJ, USA).

## In vitro transcription

Ribo RNAMax-T7 Biotin RNA labeling kit (Ribobio, Guangzhou, China) was used for the *in vitro* transcription experiment. The standard protocol was followed as per the manufacturer's guidelines. T7 RNA polymerase was used to transcribe LINC00106, with biotin used for its labeling. The antisense chain of LINC00106 was set as a negative control for this experiment.

## Western blot assay

Cell lysis buffer was used to lyse the cells (Beyotime). The total protein was extracted and separated by using Sodium dodecyl sulfate-polyacrylamide gel electrophoresis, and the separated protein was then transferred to a polyvinylidene difluoride membrane (Sigma). Bovine serum albumin (5%) was used to block the membranes that were incubated with the primary antibodies at 4 °C overnight. The following day, membranes were exposed to horseradish peroxidase-conjugated secondary antibody for 1h at room temperature. ECL-PLUS/Kit (Millipore, Bedford, MA, USA) was utilized to visualize and quantify the band signals.  Antibodies, including the anti-RPS19BP1 antibody (1:1000, ab201091) and the anti-tubulin antibody (1:5000, AT819; Beyotime), were purchased from Abcam.

## RNA-binding protein immunoprecipitation (RIP)

Magna RIP kit (Millipore, Burlington, MA, USA) was purchased and used for the RIP assay. RIP lysis buffer was used for the lysis of DU145 and PC3 cells. The cell extracts were incubated with magnetic beads coupled with specific antibodies or IgG at 4 °C for 12 h. After that, magnetic beads were washed and treated with proteinase-k to eliminate all proteins. Finally, purified RNA was subjected to RT-qPCR analysis to estimate the interactions between RPS19BP1 and LINC00106.

## RNA pull-down assay

The RNA pull-down assay was performed using the Pierce Magnetic RNA-protein pull-down test kit per the manufacturer's protocols (Thermo Scientific, Waltham, MA, USA). The RNAs were labeled with biotin using *in vitro* transcription and incubated with cell protein lysates to form an RNA-protein complex, then separated by magnetic bead binding method. Western blot analysis was conducted to detect the specific protein interaction with RNA.

## Dual-luciferase reporter assay

For the p53 activity investigation, we co-transfected the p53-Luc reporter plasmid, pcDNA3.1 expression plasmid, and siRNA into DU145 and PC3 cells for 48 h. The dual-luciferase reporter assay apparatus (Promega, Madison, WI, USA) detected reporter gene activity. Results were standardized following renal luciferase luminescence intensity (Promega-Glomax).

## Animals

The Animal Center of Soochow University supplied male BALB/C nude mice fed in a sterile environment. Feeding environment: adopt a non-toxic plastic mouse box, stainless steel wire cage cover, and metal cage, and keep the temperature of $18-29$ °C and the relative humidity of 40–70%; Feeding management: the feed intake is generally 3–7 g/day, and the feed is added 3–4 times per week; The water is supplied by drinking water bottle, the drinking water volume is generally 4–7 ml/day, and the water is changed 2–3 times a week. For *in vivo* experiment, sh-LINC00106 or an empty vector (GenePharma, Shanghai, China) was used to statically transfect PC3 cells. Each mouse received a subcutaneous injection of the transfected cells $(1 \times 10^7)$. Tumor growth was quantified every third day, and the volume of the tumor was determined with the help of the following formula: $V = 0.5 \times D \times d^2$ (V, volume; D, maximum diameter; d, maximum transverse diameter). Following 15 days of exposure, the euthanization of mice by intraperitoneal injection of three times the anesthetic dose of barbiturates was performed to record the weight and volume of the tumor The Guidelines for the Care and Use of Experimental Animals of the National Institutes of Health (NIH) and Soochow University guidelines for the euthanasia of laboratory animals were strictly followed. Ethical approval was granted from Soochow University's Animal Ethics Committee (the approval reference number is KY2022-021-01).

## Statistical analysis

Exact n numbers, statistical tests, and *P*-values were stated in figures and figure legends. All the experimental data are expressed as the mean ± standard deviation. Student's *t*-test or one-way ANOVA was performed to assess differences. In order to carry out survival analysis, the Kaplan–Meier (KM) method was utilized, while the significance analysis was performed using the log-rank test. For statistical significance, a *P*-value of $< 0.05$ was specified.

# RESULTS

## High expression of LINC00106 in human PCa tissues

The RNA data of 51 normal prostate tissues and 481 PCa tissues retrieved from TCGA suggests that LINC00106 was highly expressed in PCa tissues compared to normal tissues (Fig. 1A). Moreover, the LINC00106 level of expression significantly varied according to the stage of cancer. The patients with stage T3/T4 PCa had a significantly higher LINC00106 expression level in the affected tissues than those with stage T1/T2 prostate cancer (Fig. 1B). The disease-specific survival (DSS), overall survival (OS), and progression-free inter-period (PFI) curves were investigated to evaluate the correlation between LINC00106 expression

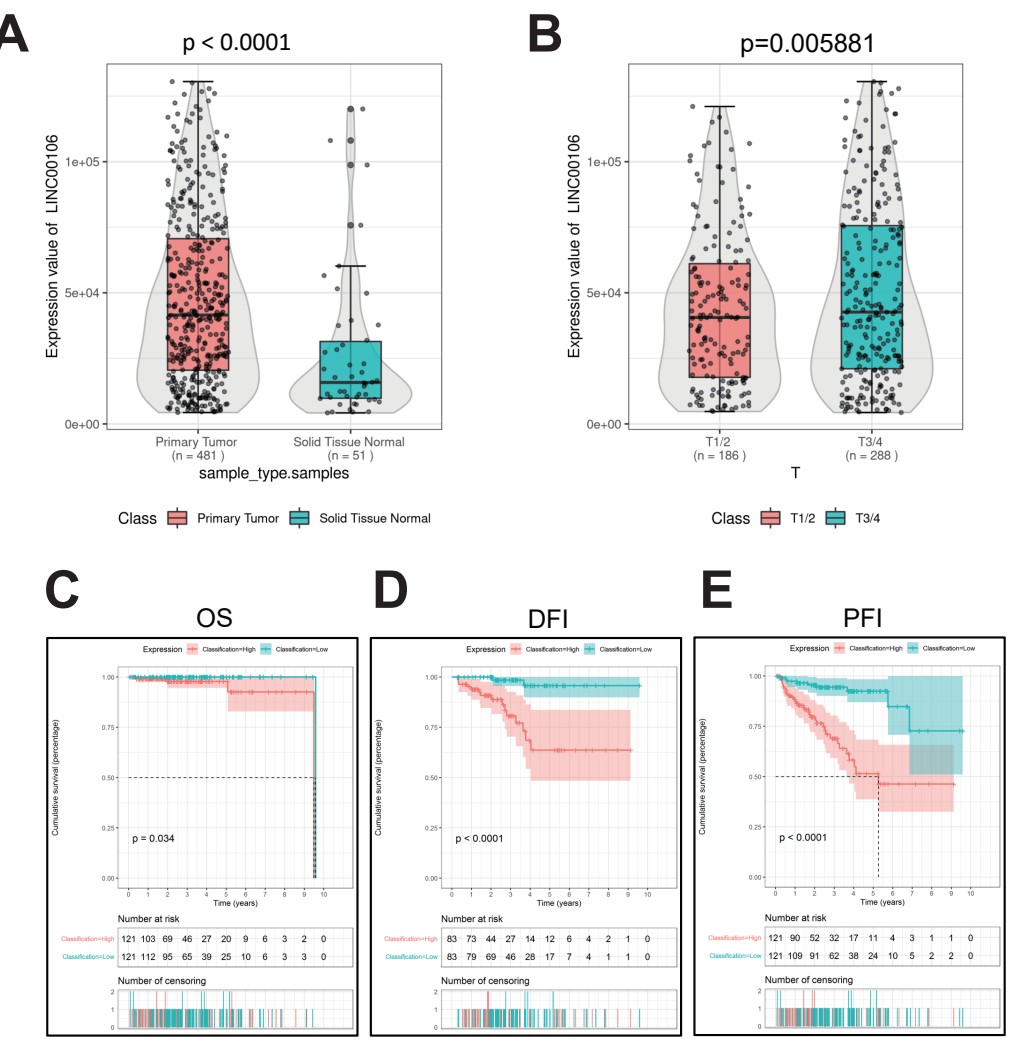

**Figure 1** **Relative expression and clinical significance of LINC00106 in PCa tissues.** (A, B) Relative expression of LINC00106 based on TCGA dataset. (C–E) Survival curves for overall survival, disease-free interval, and progression-free interval in patients with PCa.

and the prognosis of prostate cancer patients. It was revealed that high expression of LINC00106 was associated with an unfavorable prognosis among prostate cancer patients (Figs. 1C–1E).

## LINC00106 knockdown inhibits the growth of PCa *in vitro* and *in vivo*

To probe into the functions of LINC00106 in PCa cancer, we knocked down the expression of LINC00106 in DU145 and PC3 cells *via* transfecting two independent siRNAs (si-LINC00106 1# and si-LINC00106 2#, respectively). This substantially reduced the expression of LINC00106 in DU145 and PC3 cells (Figs. 2A and 2B). According to the CCK-8 assay, LINC00106 downregulation attenuated the proliferation of DU145 and PC3 cells (Figs. 2C and 2D). The Transwell experiment further established that LINC00106

downregulation attenuated the ability of DU145 and PC3 cells to migrate (Figs. 2E and 2G). Colony formation analysis revealed that the number of colonies was reduced when LINC00106 was knocked down, which is consistent with the results of the CCK8 assay (Figs. 2F and 2H). Next, to observe if LINC00106 impacted PCa cell carcinogenesis *in vivo*, PC3 cells stably expressing sh-LINC00106 or empty vector (control) were introduced into nude mice through injection. After fifteen days of injection, cells stably expressing sh-LINC00106 produced lighter and smaller tumors than cells transfected with empty vectors (Figs. 2I–2K). Immunofluorescence staining of fewer Ki67-positive cells in tumor tissues collected from the sh-LINC00106 knockdown group suggested that LINC00106 knockdown suppressed the proliferation of prostate cancer cells *in vivo* (Figs. 2L and 2M). Both *in vitro* and *in vivo* findings demonstrate that LINC00106 knockdown suppressed PCa cell proliferation and migration.

## LINC00106 interacts with RPS19BP1 in PCa cells

CatRAPID omics v2.1 is a website that predicts lncRNAs interactions with proteins. Each RNA-protein pair was scored using matrix multiplication by encoding RNA and protein sequences into the vector. RPS19BP1 had the highest interaction score with LINC00106; therefore, it was selected for further verification (Fig. 3A). Compared to normal tissues, results from the TCGA database illustrated that a remarkably higher RPS19BP1 expression level was detected in PCa tissues, consistent with LINC00106 expression (Fig. 3B). The LINC00106-RPS19BP1 interaction was confirmed by RIP assay using RNA from DU145 and PC3 cell extracts (Figs. 3C–3E). In addition, we downregulated the expression of RPS19BP1 in DU145 and PC3 cells by transfecting two independent siRNA (si-RPS19BP1 1# and 2#, respectively). The RPS19BP1 expression level was reduced significantly by both siRNAs (Figs. 3F and 3G). CCK-8 assay revealed that RPS18BP1 downregulation reduced the proliferation of DU145 and PC3 cells (Figs. 3H and 3I). Colony formation experiments showed that RPS19BP1 inhibited the viability of PCa cells (Figs. 3J and 3L). Results from the Transwell assay indicated that the migration of DU145 and PC3 cells was considerably inhibited due to RPS19BP1 knockdown (Figs. 3K and 3M). P53 is a major tumor suppressor protein involved in apoptosis. It is a nuclear transcription factor that regulates the gene expression related to senescence, growth arrest, and apoptosis in response to genotoxicity or cellular stress. However, whether RPS19BP1 can also enhance the proliferation of PCa cells by inhibiting p53 remains unclear. Therefore, a dual-luciferase reporter assay was employed to evaluate the regulatory activity of RPS19BP1 on the p53 signaling pathway in PCa cells. It was observed that transfection with si-RPS19BP1 and si-LINC00106 greatly promoted the p53 activity in both cell lines (Figs. 3N and 3O). These findings suggested that RPS19BP1 mediated the p53 signaling pathway to control the progression of PCa.

## LINC00106 interacts with RPS19BP1 to regulate p53 activity

We further explored the relationship between LINC00106 and RPS19BP1/p53 signaling pathway in PCa cells. First of all, RPS19BP1 expression was detected in the LINC00106 knockdown group. Western blot assay revealed that expression of RPS19BP1 was not altered in DU145 and PC3 cells with LINC00106 knockdown, indicating that the stability of

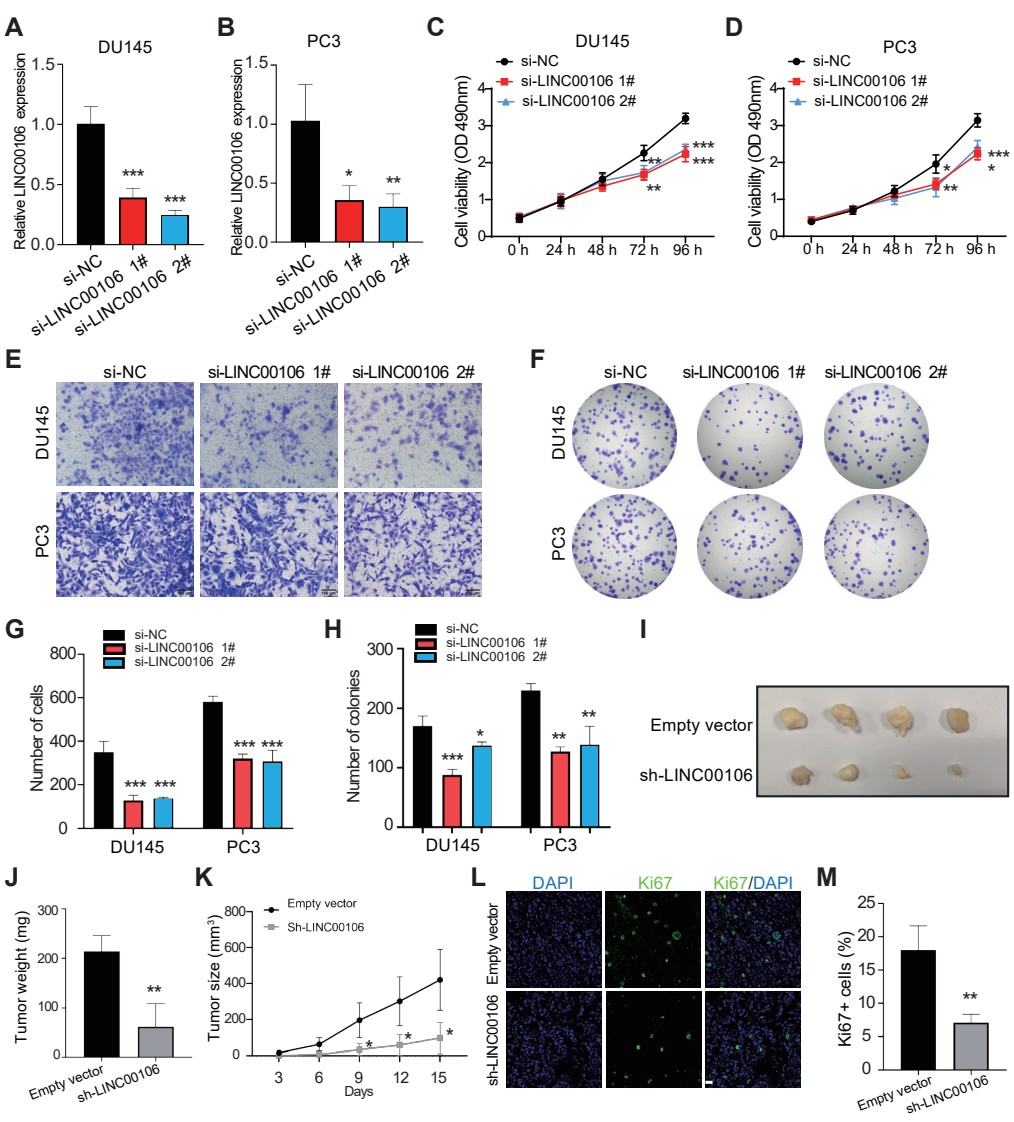

**Figure 2** The effect of LINC00106 knockdown on the viability and migration of prostate cancer cells.
(A, B) Expression of LINC00106 in DU145 and PC3 cells 48 h after transfection of negative control siRNA
(si-NC) or siRNA targeting LINC00106 (si-LINC00106 1# and 2#) in each group. (C, D) Cell viability
of DU145 and PC3 cells transfected with si-NC or si-LINC00106 1# and 2# were evaluated by CCK-8
method at 48 h. $n = 6$ for each group. (E, G) Transwell method was used to study the migration ability
of DU145 and PC3 cells in each group after LINC00106 gene knockdown for 48 h. (F, H) colony forma-
tion experiment was performed in each group to determine the proliferation capacity of DU145 and PC3
cells transfected with si-LINC00106 within 48 h. Scale bar = 100 mm. (I-K) PC3 cells stably expressing sh-
LINC00106 or empty vector (control) were injected into nude mice. (I) Tumors were isolated from nude
mice and photographed. (J) The weight of tumor. (K) Tumor volume was calculated every 3 days after in-
jection. $n = 5$ for each group. (L, M) Tumor sections were immunostained with proliferation index factor
Ki67. $n = 3$ for each group. Scale bar =50 mm. * $P < 0.05$, ** $P < 0.01$, *** $P < 0.001$.

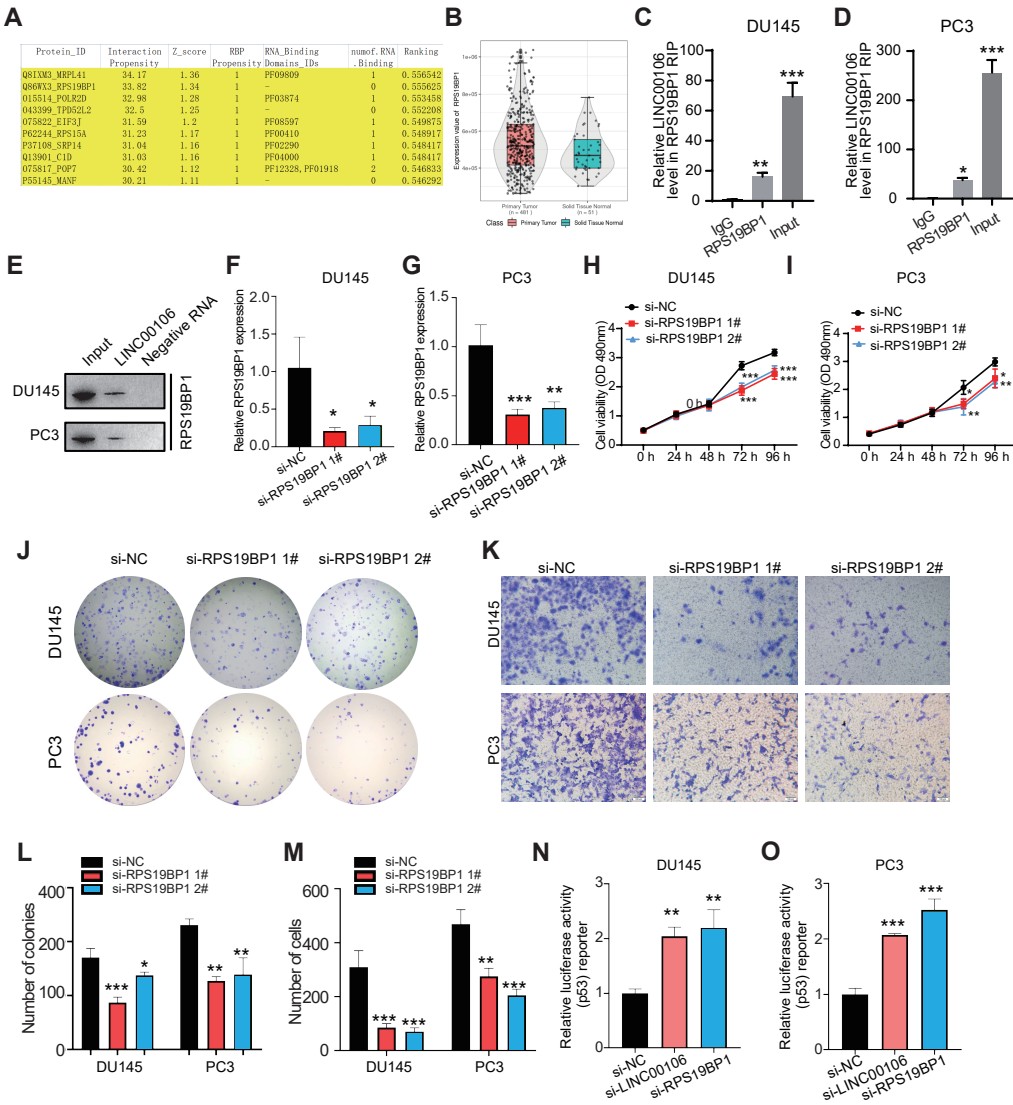

**Figure 3  LINC00106 interacts with RPS19BP1.** (A) Proteins that may interact with LINC00106 were predicted based on catRAPID omics v2.1 prediction software. (B) Relative expression of RPS19BP1 based on TCGA. (C, D) RIP assay determination of LINC00106 - RPS19BP1 binding RNA from DU145 and PC3 cell lysates. $n = 3$ for each group. (E) Biotinylated LINC00106 RNAs were incubated with DU145 or PC3 cell lysates. The RNA protein complex was analyzed by Western blotting with anti-RPS19BP1 antibody. LINC00106 antisense chain was used as negative control. (F, G) Expression of RPS19BP1 in DU145 and PC3 cells 48 h after transfection of negative control siRNA (si-NC) or RPS19BP1 targeting siRNA (si-RPS19BP1 1# and 2#) in each group. (H, I) Cell viability of DU145 and PC3 cells transfected with si-NC or si- 1# and 2# were evaluated by CCK-8 method at 48 h. (J, L) Colony formation experiments were performed in each group to determine the proliferation capacity of DU145 and PC3 cells transfected with si-RPS19BP1 within 48 h. (K, M) Transwell assay was used to study the migration ability of DU145 and PC3 cells in each group after RPS19BP1 gene knockdown for 48 h. (N, O) The P53 activity of DU145 and PC3 cells transfected with si-NC or si-RPS19BP1 1# and 2# was determined by Dual-luciferase reporter assay at 48 h. $n = 3$ for each group.

RPS19BP1 protein was not affected by LINC00106 (Figs. 4A and 4B). Then the expression of P53 was detected in LINC00106 and RPS19BP1 knockdown groups. Western blot analysis showed that the expression of P53 did not change in DU145 and PC3 cells after LINC00106 and RPS19BP1 knockdown, indicating that LIN00106 and RPS19BP1 did not change the protein level of P53 (Figs. 4C and 4D). Next, the mRNA levels of pcDNA3.1-RPS19BP1 and pcDNA3.1-RPS19BP1+si-LINC00106 obtained by PCR were significantly increased, but compared with cells treated with RPS19BP1 expression vector alone, LINC00106 knockdown did not change significantly in mRNA levels (Figs. 4E and 4F). Finally, LINC00106 knockdown significantly increased p53 activity, and RPS19BP1 was overexpressed in DU145 and PC3 cells compared with cells treated with the RPS19BP1 expression vector alone (Figs. 4G and 4H). The results confirmed that LINC00106 and RPS19BP1 did not change the protein level of P53. However, they played a role in the change of P53 activity. Therefore, LINC00106 played an important role in activating the p53 signal in PCa cells mediated by RPS19BP1 and showed that LINC00106 and RPS19BP1 together regulated the activity of p53 in PCa cells.

## DISCUSSION

LncRNAs are powerful, flexible, and pervasive tumor pathophysiological regulators important in numerous cancers, including PCa (*Hu et al., 2021*; *Hua, Chen & He, 2019*; *Hua et al., 2018*). LncRNAs exhibit complex biological functions, including acting as a miRNA sponge (*Alkan & Akgul, 2022*), a decoy for binding competitive regulatory proteins (*Qian et al., 2016*), and scaffolding or guiding for regulatory proteins or protein-DNA interactions (*Lee et al., 2016*). However, the molecular mechanism of lncRNAs regulating prostate tumorigenesis is still unclear. According to the results of this study, LINC00106 was upregulated in human PCa tissues and cells, and its overexpression was linked to an unfavorable prognosis of PCa. *In vitro* and *in vivo* studies exhibited that LINC00106 knockdown inhibited the growth of PCa cells and activated the p53 signaling pathway, and facilitated PCa progression by interacting with RPS19BP1.

LINC00106 is up-regulated in HCC tissues concerning normal liver tissues. Its level is directly associated with tumor stage, lymphatic metastasis, and OS in HCC patients (*Liang et al., 2021*). METTL3 specifically maintains LINC00106 stability by promoting the differentiation of m6A in liver cancer cells, increasing the abundance of LINC00106 in the nucleus, and promoting the stem cell and metastasis characteristics of liver cancer (*Liang et al., 2021*). In this study, RIP analysis confirmed that LINC00106 could directly interact with RPS19BP1, and its silencing greatly inhibited the survival and migration of PCa cells. Still, its functional role in PCa has not been studied.

According to previous studies, RPS19BP1 is an active regulator of SIRT1 (silencing information regulator 2 related enzymes 1) (*Knight, Willis & Milner, 2013*; *Kim et al., 2007*). SIRT1, a protein deacetylase, controls the transcription of numerous target substrates, including p53, and is a key tumor suppressor and epigenetic regulator (*Ong & Ramasamy, 2018a*; *Yi & Luo, 2010*; *Ong & Ramasamy, 2018b*). Six additional SIRT1 species have been discovered, each with distinct roles and localizations in mammals. SIRT3, SIRT4, and SIRT5

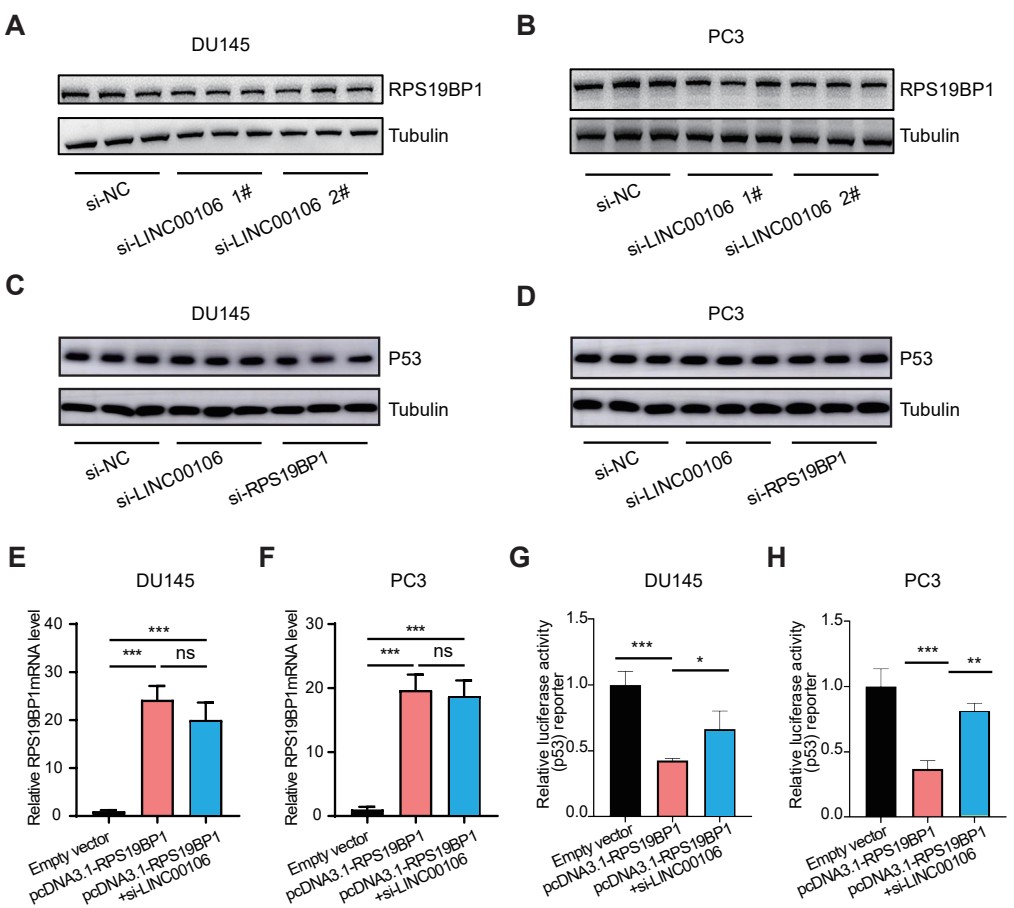

**Figure 4  LINC00106 regulates p53 activity by interacting with RPS19BP1.** (A, B) Western blotting analysis of the expression of RPS19BP1 in DU145 and PC3 cells after LINC00106 down-regulation. $n = 3$ for each group. (C, D) Western blot analysis of P53 expression in DU145 and PC3 cells after down-regulation of LINC00106 and RPS19BP1, $n = 3$ for each group. (E, F) RPS19BP1 mRNA level in DU145 and PC3 cells 48 h after transfection of empty vector or siRNA targeting RPS19BP1 (pcDNA3.1-RPS19BP1, pcDNA3.1-RPS19BP1+si-LINC001006) in each group, $n = 3$ for each group. (G, H) Dual-luciferase reporter assay was used to detect p53 activity in DU145 and PC3 cells transfected with siRNA and pcDNA3.1 vectors at 48 h per group, $n = 3$ for each group. * $P < 0.05$, ** $P < 0.01$, *** $P < 0.001$. ns: no significant.

are present in mitochondria, and SIRT2 is found in the cytoplasm, while SIRT1, SIRT6, and SIRT7 are primarily located in the nucleus (*Iside et al., 2020*; *Gonfloni et al., 2014*). SIRT1 can activate various anti-apoptotic pathways, inhibit the cell cycle, and enhance tumor growth (*Knight & Milner, 2012*). It bypasses p53-dependent cell cycle arrest and apoptosis by inhibiting the trans-activation of p53 and its downstream target genes (*Lin et al., 2020*). SIRT1 expression was not upregulated but downregulated in PCa (*Ippolito et al., 2019*), bladder cancer (*Nikas, Paschou & Ryu, 2020*), and ovarian cancer cells (*Frazzi, 2018*). It may be involved in the DNA repair pathway and double-strand break repair.

The p53 protein is almost universally recognized as a major barrier against the development and progression of most cancers (*Hassin & Oren, 2022*; *Levine, 2020*). P53 was

found to be impaired in the autophagy ability of prostate and ovarian cancer, indicating that P53 inhibited the progression of PCa (*Jin, 2005*). Therefore, in previous studies, it was found that RPS19BP1 protein could combine with SIRT1, thereby promoting the deacetylation of SIRT1 and P53 protein, which led to the loss of P53 protein function. Now, through this experiment, it is considered that LINC00106 and RPS19BP1 can further cause P53 activity changes through SIRT1 after an interaction, thus affecting the development of prostate cancer. Therefore, we believe that the inhibition of the LINC00106/RPS19BP1 axis on p53 signal transduction may promote the progress of PCa.

## CONCLUSION

We found that LINC00106 was substantially expressed in PCa patients and a significant prognostic factor in PCa patients. In addition, our results showed that a co-regulatory axis was formed between LINC00106 and RPS19BP1 that inhibits p53 activity, which promotes the proliferation and migration of PCa cells. Our data provide compelling evidence that LINC00106 promotes PCa progression and has the potential as a novel PCa biomarker.

## ACKNOWLEDGEMENTS

We thank Bullet Edits Limited for the linguistic editing and proofreading of the manuscript.

### Funding
The authors received no funding for this work.

### Competing Interests
The authors declare there are no competing interests.

### Author Contributions
- Lingxiang Lu conceived and designed the experiments, performed the experiments, analyzed the data, prepared figures and/or tables, authored or reviewed drafts of the article, and approved the final draft.
- Zhen Tian conceived and designed the experiments, performed the experiments, analyzed the data, authored or reviewed drafts of the article, and approved the final draft.
- Jicheng Lu conceived and designed the experiments, performed the experiments, prepared figures and/or tables, and approved the final draft.
- Minjun Jiang analyzed the data, authored or reviewed drafts of the article, and approved the final draft.
- Jianchun Chen analyzed the data, authored or reviewed drafts of the article, and approved the final draft.
- Shuai Guo conceived and designed the experiments, performed the experiments, analyzed the data, prepared figures and/or tables, authored or reviewed drafts of the article, and approved the final draft.

- Yuhua Huang analyzed the data, authored or reviewed drafts of the article, and approved the final draft.

### Animal Ethics

The following information was supplied relating to ethical approvals (i.e., approving body and any reference numbers):

The ethics committee of Suzhou Ninth People's Hospital provided full approval for this research (KY2022-021-01).

### Data Availability

The 18 TCGA Prostate Cancer (PRAD) datasets are available at TCGA (https://xenabrowser.net/datapages/) and the raw data are available in the Supplemental Files.

### Supplemental Information

Supplemental information for this article can be found online at http://dx.doi.org/10.7717/peerj.15232#supplemental-information.

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
