# Peer review of "LINC00106/RPS19BP1/p53 axis promotes the proliferation and migration of human prostate cancer cells"

_PeerJ, doi:10.7717/peerj.15232_

## Round 0.1 · original submission · Major Revisions

I do believe that the reviewers had several important questions that need to be addressed prior to this work being published.

Reviewer 1 ·

Basic reporting

In this study, the authors report on the interaction between LINC00106 and RPS19BP1, and their potential role in prostate cancer cells. They demonstrate that LINC00106 and RPS19BP1 are important for the proliferation and migration of the cells. Mechanistically, they propose that LINC00106 and RPS19BP1 act through regulating the level of p53.

Overall, I find the phenotypes of LINC00106/RPS19BP1 knockdown clear. However, the study lacks mechanistic insights, and therefore needs some major revision.
1. Importantly, the authors did not examine the RNA and protein level of p53 upon knockdown of LINC00106 and RPS19BP1. Therefore, it is not clear whether the effect of LINC00106/RPS19BP1 is through p53.
2. If the expression of p53 is indeed changed upon knockdown of LINC00106/RPS19BP1, the authors need to demonstrate whether/how LINC00106/RPS19BP1 regulates the level of p53 directly or indirectly by inducing p53-dependent cell death.

Without these experiments, the role of p53 is really unclear in the story.

Experimental design

1. In figure 1B, the difference between the two groups is very minimal. Therefore, the relevance to the story is questionable.
2. In figure 1C-E, it is unclear how the authors separate the high/low expression of LINC00106.
3. With the luciferase assay, a proper epistasis analysis should be performed, preferentially with a clean knockout system. The data in figure 4c-d are not informative.

Validity of the findings

No comment

Additional comments

Somehow, the figures in the version given to the reviewer has very low resolution.

Reviewer 2 ·

Basic reporting

1 The review of related work is not sufficiently thorough. The authors should cite references if appropriate. For example, according to L18-L19, L39-L41, L58-L59, and L63-L64, the authors didn’t follow up with the recent LINC00101 or PCa research works such as https://cancerci.biomedcentral.com/articles/10.1186/s12935-020-01577-1.
2 L55-L57 seems not a correct statement.
3 Many references are missing throughout the manuscript, such as L73-L74.
4 The authors should improve some statements, for example, L68-L69 and L138-L139 are lacking in logic.
5 I don’t think Figure 3A is good. The authors should remake it.
6 I suppose “Figure 3C and D” is not correct in L214.
7 ‘gene expression’ seems not appropriate in L246 and L91.
8 “Insulin-like growth factor 2 mRNA-binding protein 1” is weird and should be replaced.
9 The authors should note punctuation marks and spaces throughout the manuscript, such as L60.
10 The organization of supplemental files is terrible. Also. It’s very hard for readers to understand these files.

Experimental design

1 The authors should add more details in Materials & Methods.
2 How to define the high expression of LINC00106 in L182-L183?
3 “LncRNAs with an expression level greater than 0.3 in all samples were screened” in L81-L82. Is there any reference or reason to support this processing step?

Validity of the findings

1 “Sequencing data” is used throughout the manuscript, but it’s not specific at all…
2 ‘clinical genomics and proteomics data were downloaded’ in L82-lL84, again, it is not clear what kind of data was downloaded.
3 The authors should provide a sample size, test (test method, two-sided or one-side?), and p-value in the manuscript for all statistical tests they did. Also, it’s not clear for example, what n=6(or other numbers) means in Figure 2C and 2D, the authors should clarify them.

Additional comments

1 I suggest that the Discussion section should be improved to better reflect the quality of the work.

·

Basic reporting

NA

Experimental design

NA

Validity of the findings

NA

Additional comments

Minor Comments

The manuscript entitled “LINC00106/RPS19BP1/p53 axis promotes the proliferation and migration of human prostate cancer cells” is a well written and is an original work. This work is well under the domain of this journal. The authors have addressed an important topic of research having very important clinical relevance. This study investigates the role of long non-coding RNA (LINC00106) in prostate cancer and its potential as a biomarker candidate in clinical investigations.
I recommend this work to be published in this journal. However, before that I have few minor comments which need to be addressed before that:
1. Reference are missing at some places, authors should provide the relevant reference throughout the manuscript wherever missing. Few examples are at line number 40-41, 248-251 etc.
2. In material method section, typo error: primer sequence AGgACC……..
3. In line 259, please delete “in addition to SIRI1” as it is redundant in the sentence. Please see line number: 275.
4. The image quality of few figures is poor. Authors should improve the image quality of figure 3A, Figure 3A &B. The labels are hardly readable.
5. In line 160, the authors talk about how they calculated tumor volume but have not mentioned how the tumor was visualized.
6. In line 176 “normal tissues” is mentioned, authors should make it clear whether it is prostate tissue or any other tissues. If other tissues, then please specify.
7. Legend of figure 1 should mention “tissues” only instead of “tissue and cell lines”.
8. The first line of figure’s legend is duplicate of the figure tittle, which make it redundant. Authors should replace it if possible.
9. The discussion section of the manuscript should be more elaborate, discussing the important findings of the work in more detail. Authors should try to improve it with discussing more relevant studies and their finding which supports or contradict with the outcomes of the present work. For example, authors should discuss the pattern of expression of LINC00106 in various other tumors. How in thyroid cancer the expression of said lncRNA is downregulated and contrary to that, its expression in prostate cancer is upregulated as reported in this study? Authors should also try to discuss this difference with possible explanations.

---

## Round 0.2 · Minor Revisions

Reviewer 2 had several questions that need to be addressed prior to this work being published.

Reviewer 1 ·

Basic reporting

The authors have successfully addressed all my comments. Therefore, I recommend publication of this work.

Experimental design

No comment

Validity of the findings

No comment

Additional comments

No comment

Reviewer 2 ·

Basic reporting

1 The authors should improve some statements, for example, L68-L69 and L138-L139 are lacking in logic.
2 I don’t think Figure 3A is good. The authors should remake it.
3 The organization of supplemental files is terrible. Also. It’s very hard for readers to understand these files.

Experimental design

1 The authors should add more details in Materials & Methods.
2 How to define the high expression of LINC00106 in L182-L183?
3 “LncRNAs with an expression level greater than 0.3 in all samples were screened” in L81-L82. Is there any reference or reason to support this processing step?

Validity of the findings

1 “Sequencing data” is used throughout the manuscript, but it’s not specific at all…
2 ‘clinical genomics and proteomics data were downloaded’ in L82-lL84, again, it is not clear what kind of data was downloaded.
3 The authors should provide a sample size, test (test method, two-sided or one-side?), and p-value in the manuscript for all statistical tests they did. Also, it’s not clear for example, what n=6(or other numbers) means in Figure 2C and 2D, the authors should clarify them.

Additional comments

Although the author made some revisions, I don't think they took these comments seriously, and they didn't address them either. Please take time and think about how to make the manuscript better!

·

Basic reporting

na

Experimental design

na

Validity of the findings

na

Additional comments

The authors have addressed all the commnets/questions in this revised manuscript. Therefore, I recommned this revised manuscript to be published in this journal.

---

## Round 0.3 · Minor Revisions

The manuscript is almost ready to be accepted for publication in PeerJ but it requires editing for language and typographical errors.

---

## Round 0.4 · accepted · Accept

I recommend publishing in this present form.